# The Approach of Artificial Intelligence in Neuroendocrine Carcinomas of the Breast: A Next Step towards Precision Pathology?—A Case Report and Review of the Literature

**DOI:** 10.3390/medicina59040672

**Published:** 2023-03-28

**Authors:** Diana Maria Chiorean, Melinda-Ildiko Mitranovici, Maria Cezara Mureșan, Corneliu-Florin Buicu, Raluca Moraru, Liviu Moraru, Titiana Cornelia Cotoi, Ovidiu Simion Cotoi, Adrian Apostol, Sabin Gligore Turdean, Claudiu Mărginean, Ion Petre, Ioan Emilian Oală, Zsuzsanna Simon-Szabo, Viviana Ivan, Ancuța Noela Roșca, Havva Serap Toru

**Affiliations:** 1Department of Pathology, County Clinical Hospital of Targu Mures, 540072 Targu Mures, Romania; 2Department of Obstetrics and Gynecology, Emergency County Hospital Hunedoara, 14 Victoriei Street, 331057 Hunedoara, Romania; 3Department of Obstetrics and Gynecology, ”Victor Babes” University of Medicine and Pharmacy, 2 Eftimie Murgu Sq., 300041 Timisoara, Romania; 4Public Health and Management Department, ”George Emil Palade” University of Medicine, Pharmacy, Science and Technology of Targu Mures, 540139 Targu Mures, Romania; 5Faculty of Medicine, “George Emil Palade” University of Medicine, Pharmacy, Sciences and Technology, 540142 Targu Mures, Romania; 6Department of Anatomy, ”George Emil Palade” University of Medicine, Pharmacy, Sciences and Technology, 540142 Targu Mures, Romania; 7Department of Pharmaceutical Technology, ”George Emil Palade” University of Medicine, Pharmacy, Sciences and Technology, 540142 Targu Mures, Romania; 8Close Circuit Pharmacy of County Clinical Hospital of Targu Mures, 540072 Targu Mures, Romania; 9Department of Pathophysiology, ”George Emil Palade” University of Medicine, Pharmacy, Science, and Technology of Targu Mures, 38 Gheorghe Marinescu Street, 540142 Targu Mures, Romania; 10Department of Cardiology, “Victor Babes” University of Medicine and Pharmacy, 2 Eftimie Murgu Sq., 300041 Timisoara, Romania; 11Department of Obstetrics and Gynecology, “George Emil Palade” University of Medicine, Pharmacy, Sciences and Technology, 540142 Targu Mures, Romania; 12Department of Medical Informatics and Biostatistics, “Victor Babes” University of Medicine and Pharmacy, 2 Eftimie Murgu Sq., 300041 Timisoara, Romania; 13Department of Cardiology, ”Pius Brinzeu” County Hospital, 2 Eftimie Murgu Sq., 300041 Timisoara, Romania; 14Department of Surgery, ”George Emil Palade” University of Medicine, Pharmacy, Sciences and Technology, 540142 Targu Mures, Romania; 15Department of Pathology, Akdeniz University School of Medicine, Antalya Pınarbaşı, Konyaaltı, 07070 Antalya, Turkey

**Keywords:** histopathology, whole slide image (WSI), explainable artificial intelligence (AI), breast neuroendocrine tumors (NETs), early breast carcinoma, pattern recognition, deep learning

## Abstract

Primary neuroendocrine tumors (NETs) of the breast are considered a rare and undervalued subtype of breast carcinoma that occur mainly in postmenopausal women and are graded as G1 or G2 NETs or an invasive neuroendocrine carcinoma (NEC) (small cell or large cell). To establish a final diagnosis of breast carcinoma with neuroendocrine differentiation, it is essential to perform an immunohistochemical profile of the tumor, using antibodies against synaptophysin or chromogranin, as well as the MIB-1 proliferation index, one of the most controversial markers in breast pathology regarding its methodology in current clinical practice. A standardization error between institutions and pathologists regarding the evaluation of the MIB-1 proliferation index is present. Another challenge refers to the counting process of MIB-1′s expressiveness, which is known as a time-consuming process. The involvement of AI (artificial intelligence) automated systems could be a solution for diagnosing early stages, as well. We present the case of a post-menopausal 79-year-old woman diagnosed with primary neuroendocrine carcinoma of the breast (NECB). The purpose of this paper is to expose the interpretation of MIB-1 expression in our patient’ s case of breast neuroendocrine carcinoma, assisted by artificial intelligence (AI) software (HALO—IndicaLabs), and to analyze the associations between MIB-1 and common histopathological parameters.

## 1. Introduction

Primary neuroendocrine tumors (NETs) of the breast represent a rare type of tumor, whose origin comes from cells producing peptides and amines, cells that make up the diffuse neuroendocrine system, whose significance remains insufficiently studied. Regarding the histogenesis, there are two main theories, according to Rosen and Gatuso. The most controversial claims that these tumors evolve through the neoplastic transformation of native neuroendocrine cells, with some authors supporting their existence in benign breast tissue, a theory that has remained, for the most part, contested and unconfirmed. The second theory, however, refers to the fact that neuroendocrine differentiation comes from a divergent differentiation of neoplastic stem cells, during the early phase of carcinogenesis, from epithelial and endocrine cells. This theory was supported, the neuroendocrine cells being clonally related to the malignant epithelium [1]. Because it can mimic some of the most common histological subtypes of breast cancer, primary neuroendocrine carcinoma of the breast remains difficult to diagnose and, therefore, underrecognized [2].

The true incidence of the disease is difficult to assess because neuroendocrine immunohistochemical markers are not routinely used in the diagnosis of breast tumors. Due to its low incidence, there is a paucity of evidence regarding the optimal management and prognostic relevance of mammary neuroendocrine tumors, the only available reference data being those resulting from case reports or case series [3].

To establish a final diagnosis of breast carcinoma with neuroendocrine differentiation, it is essential to perform a neuroendocrine immunohistochemical profile of the tumor using antibodies against synaptophysin or chromogranin, as well as the MIB-1 (mindbomb E3 ubiquitin protein ligase 1) proliferation index, one of the most controversial markers in breast pathology with regard to its methodology in current clinical practice. For the majority of this index, it is imperative to precisely measure an indispensable nuclear antigen for cell proliferation [4], as the methods of digitally analyzing the images are of great use [5]. Routinely, pathologists visually evaluate MIB-1 by manually counting cells using an optical microscope, even though this method lacks repeatability among observers [4,6,7,8] and leads to limitations in its clinical application [4,9].

Advances of technology in the digitization process of histopathological slides, a phenomenon known as digitizing glass slides, has led to a decrease in their storage prices. The whole slide images (WSIs) scan allows their users to digitally examine various specimens [10] and to develop remote consultation systems, including pathological experts worldwide, with greater ease [10,11].

## 2. Case Report

We present the case of a post-menopausal 79-year-old G3P1 (gravida 3, para 1) woman, who presented to our clinic in October 2021, alleging the presence of a left breast tumor, described as slowly growing but rapidly enlarged, 2 weeks before presentation to the hospital. The patient denied having fever, nipple discharge or a family history of breast carcinoma.

On physical examination, the left breast was tender and the overlying skin erythematous, with a firm mass felt predominantly in the upper quadrant of the breast. At the time of examination, the patient was afebrile, with stable vital signs. The patient was referred to a breast ultrasonography after a core biopsy of the mass was performed. The ultrasound examination of the left breast revealed a hypoechoic, irregular nodule, with a spiculated outline, with dimensions of approximately 15.1 × 13.5 × 18.2 mm and with posterior acoustic attenuation, intralesional vascularization and associated architectural distortion.

The histopathological examination of the core tissue on a routine histological stain (Hematoxiline and Eosine) revealed a tumor proliferation with an infiltrative appearance, consisting of plasmacytoid cells arranged in solid nests (Figure 1).

The tumor occupied approximately 90% of the length of the biopsy, with an average cellularity of approximately 35% and a TILs (tumor infiltrating lymphocytes) score of 15%.

The immunohistochemical techniques using antibodies against NSE (neuron specific enolase) (MRQ-55), Chromogranin A (LK2H10) and Synaptophysine (MRQ-40) (Rabbit Monoclonal Antibodies, CELL MARQUE, VENTANA) (Figure 2) were consistent, orienting the pathologist towards the diagnosis of an infiltrative neuroendocrine tumor (NET) of the breast, grade 2 (G2), according to Nottingham criteria (tube formation—3, nuclear pleomorphism—3, number of mitoses—1, total score = 7).

A contrast-enhanced computed tomography (CT) examination of the thorax, abdomen and pelvis revealed a breast nodule with the size of 16 × 15 mm and with a spiculated outline and an inhomogeneous contrast uptake at the level of the supero-external quadrant, without axillary adenopathy. No significant abnormalities were detected in the rest of the thorax, abdomen or pelvis, including the visualized spine bones.

In January 2022, following the decision of the therapeutic committee, the patient underwent a left radical mastectomy, for which the mammographic examination highlighted a micropolylobulated hyperdense opacity, with a size of 20 × 18 mm, homogeneous and with grouped and dispersed microcalcifications.

Subsequently, the surgical specimen was sent to the pathology department for histological and immunohistochemical evaluations.

At gross examination, the mastectomy specimen measured 185 × 130 × 55 mm, revealing a nodular and infiltrative tumoral mass of a firm consistency, whitish color, with the size of 18 × 15 × 20 mm and located at a distance of 20 mm from the upper surgical resection margin, a 90 mm distance from the lower surgical resection margin, a 20 mm distance from the anterior surgical resection margin and a 20 mm distance from the posterior surgical resection margin, identified at the level of the supero-external quadrant (Figure 3).

The microscopy on the routine histological stain (Hematoxiline and Eosine) revealed a tumor proliferation with an infiltrative aspect, consisting of plasmacytoid tumor cells arranged in solid nests—as already confirmed on the core tissue—separated by delicate conjunctival-vascular septa (Figure 4).

On some sections, fibrocystic changes were observed, represented by dilated ductal structures, lined by a flattened epithelium, in the lumen of which an acellular eosinophilic material was observed in some ducts with the presence of histiocytes, vacuolated cytoplasm and a small, round and centrally located nucleus, as well as a lesion with a disorganized lobular architecture, represented by a marked hyperplasia of ductal structures lined by a bilayered epithelium.

The results of the immunohistochemical techniques, using antibodies against Synaptophisine, Chromogranin A and NSE (neuron specific enolase), were consistent, as well, confirming the diagnosis of invasive neuroendocrine (NET) carcinoma of the breast, grade 2 (G2). The MIB-1 proliferation index was performed with manual counting using light microscopy; it resulted in a percentage of 3%. Axillary lymph nodes did not reveal metastases, and post-operatively, the patient was treated with three cycles of adjuvant chemotherapy.

### 2.1. Artificial Intelligence Approach and Slide Digitization

After a careful examination of all the Hematoxylin and Eosin (HE)-performed sections, the most representative paraffin block containing the tumor with its largest dimensions was selected. Serial formalin-fixed paraffin embedded (FFPE) tissue sections were prepared with a 3 µm thickness.

All the sections were digitally scanned with a KFBIO KF-PRO-020 whole-slide scanner (the product of KONFOONG BIOTECH INTERNATIONAL CO. LTD.), using a single z-plane and a resolution of 0.25 micrometer/pixel at 40× magnification. The scanner was able to scan a 15 × 5 mm area in less than 100 s, with a position accuracy of less than 20 nm. The area of tissue under a single field of view (FOV) at 40× was 0.035 mm^2^, which resulted in a size of 1324 × 1096 pixels. To secure a high successful scan rate (above 99.85%), the region of interests (ROIs) and focus points in the ROI were automatically recognized by the scanner (Figure 5). In order to reach a high image quality, in case it was needed, focus points were manually updated by the user.

Cytonuclear, Area-Quant and Multiplex IHC modules of HALO (product of the company ”IndicaLabs”) were used for the quantitative analysis of the whole slide images. MIB-1 stained images were registered and viewed with the help of the registration and Figure Maker Tools of the SW.

### 2.2. Evaluation of the MIB-1 Proliferation Index and Establishment of the Histological Grade

The manual count of the expert pathologist using light microscopy with regard to the MIB-1 proliferation index (defined as hot spot counting) was used as the gold standard. To evaluate the general concordance rate of the method assisted by artificial intelligence (AI), respectively by manual counting, the MIB-1 proliferation index calculation was compared with the pathologist’s report.

The manual evaluation of the MIB-1 proliferation index was performed on 10 HPF (high power fields)/40× magnification.

The result regarding the automatic counting of the MIB-1 proliferation index correctly estimated the manual values reported by our expert pathologist—3% percentage.

Although the process of MIB-1 manual counting is subjective, and it is recommended to have it evaluated by an experienced pathologist, we would like to emphasize the efficiency demonstrated by using AI, thus facilitating the diagnosis.

## 3. Discussion

Neoplasms with neuroendocrine differentiation do not represent a specific histopathological category of the female breast carcinomas spectrum. However, they are considered capable of producing abnormal hormonal substances and, therefore, represent a subgroup of breast carcinomas whose pattern recognition is absolutely necessary for defining them clinically. They still remain an underrecognized entity due to the limited cytological evidence regarding their differentiation and the lack of a consensus regarding the pathological degree [1] for their early diagnosis, and it is necessary to exclude the metastases of the extramammary origin, which represents a real challenge for the pathologist in establishing a final diagnosis due to its variable cytological characteristics [1,12]. In order to be able to exclude a possible breast metastasis with neuroendocrine differentiation of another origin, it is recommended to perform a computed tomography (CT) scan, as in our patient’s case; somatostatin receptor (SSTR) scintigraphy; or PET-CT with gallium-labeled somatostatin analogs 68 [3,13].

In terms of therapy, a study led by Vranic et al., highlighted the predictive expression of TROP 2 (a transmembrane glycoprotein encoded by the Tacstd2 gene), FOLR1 (folate receptor 1) and H3K36Me3 (an epigenetic modification to the DNA packaging protein Histone H3) for different tumor subsets, a study that can pave the way in the development of a new targeted therapy for patients with NEBCs (primary neuroendocrine breast carcinomas) [3,14].

The term “artificial intelligence”, used for the first time in 1955 at a conference organized by Dr. John McCarthy, is meant to describe “thinking machines” [10,15].

In the 1980s, computers became more accessible to the general public under the name of personal computers. Although an attempt was made to use the knowledge of experts in this new field of AI in order to prepare machines to be used in solving problems, its scope was limited, and this initiative ended without a significant discovery [10,16].

Regarding digital pathology, this transition has been slower due to many factors, including the difficulty of assessing a return on the investment, which remains challenging [10].

Although there have been numerous discussions regarding the benefits of digitization, most pathologists have avoided the implementation of this digital practice, not wanting to change the workflow [10,17,18].

The research of Li et al., which combined AI with manual methods, involved 300 cases of breast cancer in their study, in different stages of the disease. They suggest there is a high degree of consistency regarding the mentioned methods of counting the MIB-1 proliferation index, and they also claim that the estimation of this proliferation index by the artificial intelligence method not only encounters the recommendations of the International Breast Cancer Working Group in terms of a specific number of cells, but also exceeds the lapse of the proposed cell numbers [4].

According to the results obtained by Li et al. and Stålhammar et al. [4,19], AI software has high accuracy and repeatability in the interpretation of MIB-1 immunohistochemistry.

We would like to emphasize the benefits of applying AI in pathology, mentioning standardization and reproducibility, improvement of the diagnostic accuracy, development of the subspecialty expertise, and a more increased efficiency in terms of diagnosing the early stages of breast cancer [10,20,21]. The cell proliferation index MIB-1 offers a good perspective regarding the behavior of the tumor from a biological point of view, being particularly important in early grading or subtyping certain neoplasms, such as brain tumors, breast carcinomas, non-Hodgkin’s lymphoma or neuroendocrine tumors [21,22,23].

Most of the methods used for automatic detection of MIB-1 have reported the obtained results, but these are not yet widely available for the public [24,25,26,27] due to their experimental character. Future studies regarding the utility of artificial intelligence (AI) technology in routine practice are still required.

The analysis of the above-mentioned hot spots allows the identification of the most aggressive regions of tumors with a heterogeneous proliferative character. In routine diagnosis, the percentage of MIB-1 identified manually by the pathologist involves the manual counting of a limited number of tumor cells.

The advantage of an analysis assisted by artificial intelligence is represented by the consistency of the score obtained between users, with a beginner being able to be trained to use it in an extremely short time, obtaining a concordance of the results almost identical to those obtained by an experienced pathologist [19,28,29,30,31,32]. Regarding breast cancer screening, according to Larsen et al., AI can be used in very different ways, with various AI systems being designed to be used in particular configurations [33].

Nowadays, AI represents a promising start in the care management of breast cancer patients, as we are facing a very large increase of the number of patients. Therefore, we can say that it has good potential for streamlining programming, as well as for ensuring a continuous follow-up. In the next decade, we predict that artificial intelligence will play an extremely important role in terms of establishing a personalized treatment for each individual patient, as well [34].

Our case presents a series of deficiencies that we consider necessary to be addressed in the future. This analysis being carried out in a single academic center and being reported on in a single case of neuroendocrine breast tumor (NET) raises the issue of the occurrence of irregularities from a logistical point of view, a situation that is still not quite clear. Secondly, the MIB-1 score, being manually generated by a single pathologist, cannot express in a very clear manner the measure of variability between observers.

## 4. Conclusions

Regarding the MIB-1 methodology in current clinical practice, it is imperative that it be precise as a measurement, especially due to the controversies regarding its utility in breast cancer and its early stages, as well as digital analysis of the images being very useful. This new approach that involves AI proves to be of real value, the automatic estimation of the MIB-1 proliferation index requiring a small number of training samples, samples represented by the signed reports of the expert pathologists. This evaluation proved that the prototype is promising, and we consider that it presents an increased potential that deserves to be exploited further in clinical practice, as it is a real help for pathologists who evaluate this index [23]. In our case, it was represented in a percentage of 3%, in both light microscopy—digital slide manual counting and AI-assisted IndicaLabs CytoNuclear, version 2.0.2.9′.

Not least, advances of technology in the digitization of histopathological slides are for the benefit of patients, allowing us to develop remote consultation systems between pathological experts worldwide, as well as improving diagnostic accuracy and its efficiency.

## Figures and Tables

**Figure 1 medicina-59-00672-f001:**
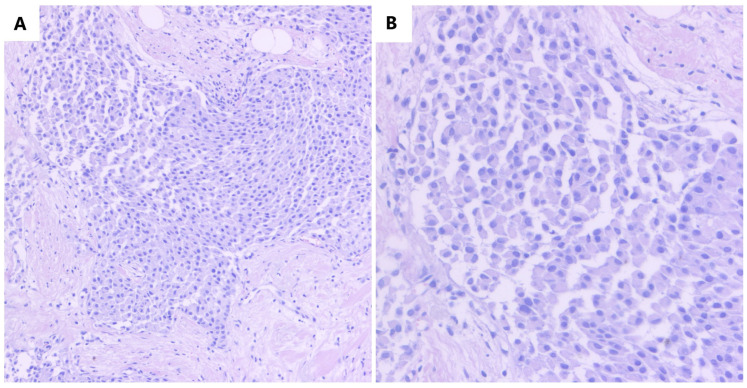
(**A**) Core tissue examination revealing a tumor proliferation with an infiltrative aspect, consisting of plasmacytoid cells arranged in solid nests (HE, ob. 10×); (**B**) Details of the described area (HE, ob. 20×).

**Figure 2 medicina-59-00672-f002:**
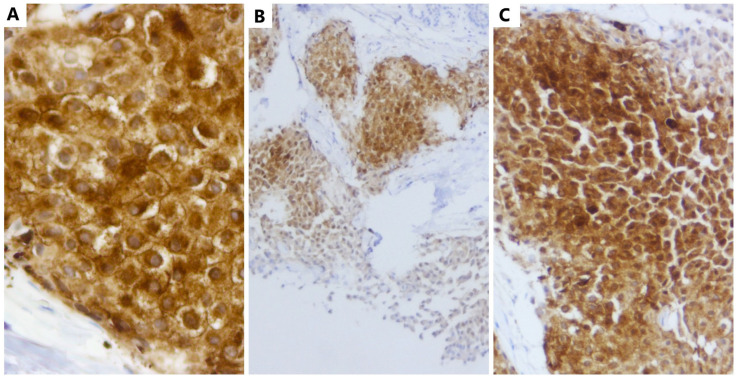
Neuroendocrine tumor of the breast: (**A**) NSE antibody (strong expression, ob.40×); (**B**) Chromogranin A antibody (strong and diffuse expression, ob.10×); (**C**) Synaptophysine antibody (strong expression, ob.20×).

**Figure 3 medicina-59-00672-f003:**
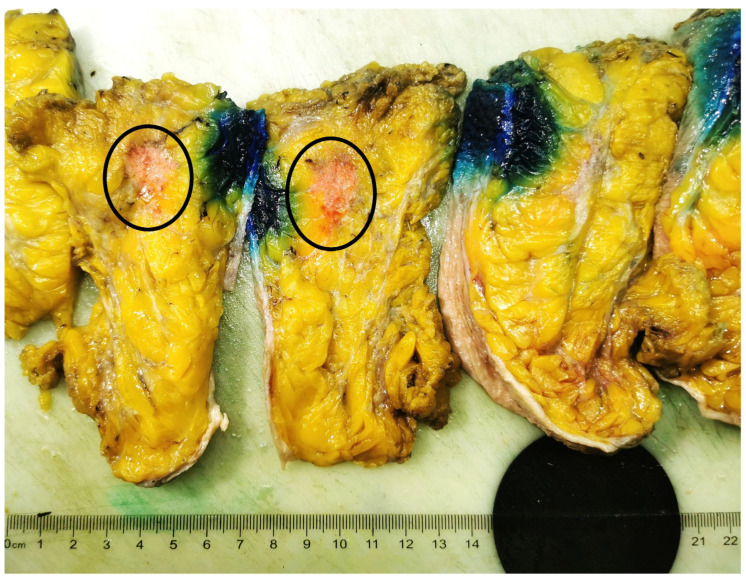
Gross examination of the mastectomy specimen revealing a nodular tumoral mass with infiltrative margins (within the black circle), located at the level of the SE (supero-external) quadrant of the breast.

**Figure 4 medicina-59-00672-f004:**
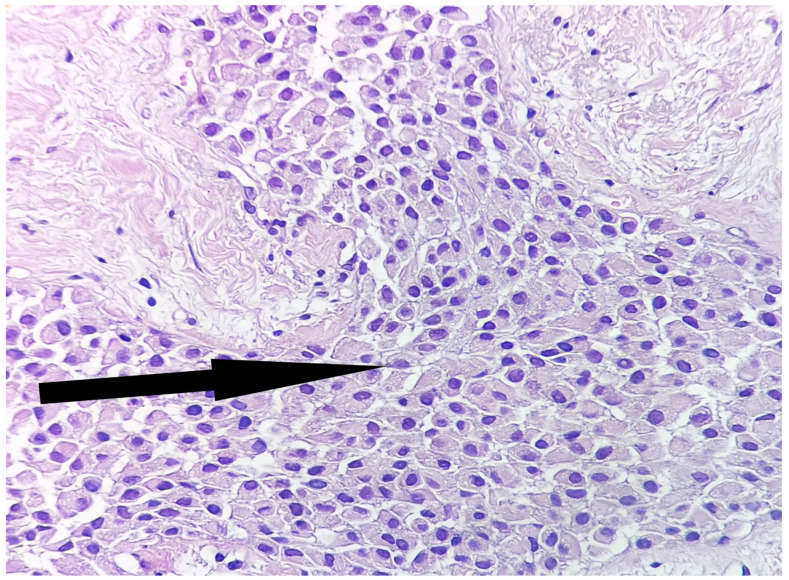
Tumor proliferation with an infiltrative aspect, consisting of plasmacytoid tumor cells, arranged in solid nests—black arrow (HE, ob.20×).

**Figure 5 medicina-59-00672-f005:**
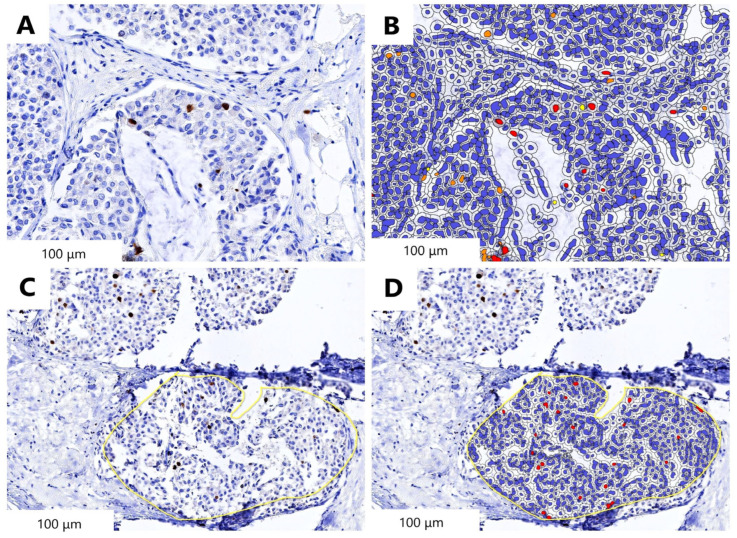
(**A**) Automatic detection of MIB-1′s hot-spot area; (**B**) Detected tumor area (blue) and immunopositive cells (red); (**C**) Manually annotated tumor area (yellow outline); (**D**) Detected tumor area (blue cells) and immunopositive cells (red) inside the annotated area (yellow outline). The length bars shown in the WSIs (whole slide images) are 100 μm.

## Data Availability

Not applicable.

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
