# Peer review of "The Approach of Artificial Intelligence in Neuroendocrine Carcinomas of the Breast: A Next Step towards Precision Pathology?—A Case Report and Review of the Literature"

_medicina, 2023, doi:10.3390/medicina59040672_

Round 1

Reviewer 1 Report

Application of ai in the field of cancer detection and diagnosis is an interesting field. I congratulate with authors for the paper. 
I suggest to add some reference in your manuscript to complete the discussion regarding the use of AI ij several phases of cancer detection (PMID: 36917260), and management ( PMID: 20336782, PMID: 36773458)

best regards

Author Response

Dear Reviewer,

Thank you very much for your helpful suggestions. We followed your advice accordingly.

The completion of references and discussions correspond to paragraphs: 284-292, 422-426.

Thank you very much for your support and kind words.

Sincerely,
Dr. Diana Maria Chiorean

Pathology resident

Pathology Department of the County Clinical Hospital of Târgu MureÈ™, Romania

Reviewer 2 Report

A case report concerning the diagnostics of neuroendocrine carcinoma in post-menopausal 79-years old patient is presented in the manuscript. The authors compared MIB-1 proliferation index obtained by the expert pathologist with the result of artificial intelligence (AI). No doubt that the fact of similarity of both results is very promising. But the manuscript depicts only one case report. It seems that many efforts should be made in future to confirm matching of the routine histopathological results with those obtained with AI assistance.

Nevertheless this consideration does not diminish the importance of the research in this area.

The results are obtained using modern methods and are well illustrated. The references are adequate and give full information about the used methods and the state-of-art in treating the problem. The conclusion is made accurately and does not pretend to be final for the evaluated aspects of the diagnostics.

There are some minor problems with English language, so moderate English changes required.

Author Response

Dear Reviewer,

Thank you very much for your helpful suggestions. We followed your advice accordingly.

We corrected some manuscript parts, corresponding to paragraphs: 64-67, 74-75, 79-83, 89, 129, 133, 151, 152, 174, 182.

Thank you very much for your support and kind words.

Sincerely,
Dr. Diana Maria Chiorean

Pathology resident

Pathology Department of the County Clinical Hospital of Târgu MureÈ™, Romania

Reviewer 3 Report

  The manuscript is about the diagnostics of neuroendocrine carcinoma in post-menopausal 79-years old patient. The authors compared MIB-1 proliferation index obtained by the expert pathologist with the result of artificial intelligence. The case report is well described and the authors take their case for a careful analysis of the literature. The topic is very new and  modern methods and are well illustrated. The references are adequate. The conclusion is made accurately. Minor English language revision is required.

Author Response

Dear Reviewer,

Thank you very much for your helpful suggestions. We followed your advice accordingly.

We corrected some parts of the manuscript, corresponding to paragraphs: 64-67, 74-75, 79-83, 89, 129, 133, 151, 152, 174, 182.

Thank you very much for your support and kind words.

Sincerely,
Dr. Diana Maria Chiorean

Pathology resident

Pathology Department of the County Clinical Hospital of Târgu MureÈ™, Romania
